# Temperature-Dependent Exciton Dynamics in a Single GaAs Quantum Ring and a Quantum Dot

**DOI:** 10.3390/nano12142331

**Published:** 2022-07-07

**Authors:** Heedae Kim, Jong Su Kim, Jin Dong Song

**Affiliations:** 1School of Semiconductor Science, Technology and Semiconductor Physics Research Center, Jeonbuk National University, Jeonju 561-756, Korea; 2Department of Physics, Yeungnam University, Gyeongsan 38541, Korea; 3Center for Opto-Electronic Convergence Systems, Korea Institute of Science and Technology, Seoul 136-791, Korea

**Keywords:** quantum ring structure, quantum dot structure, exciton, photoluminescence, localized states, polarization dependence, fine structures, strong confinement

## Abstract

Micro-photoluminescence was observed while increasing the excitation power in a single GaAs quantum ring (QR) at 4 K. Fine structures at the energy levels of the ground (*N* = 1) and excited (*N* = 2) state excitons exhibited a blue shift when excitation power increased. The excited state exciton had a strong polarization dependence that stemmed from the asymmetric localized state. According to temperature-dependence measurements, strong exciton–phonon interaction (48 meV) was observed from an excited exciton state in comparison with the weak exciton–phonon interaction (27 meV) from the ground exciton state, resulting from enhanced confinement in the excited exciton state. In addition, higher activation energy (by 20 meV) was observed for the confined electrons in a single GaAs QR, where the confinement effect was enhanced by the asymmetric ring structure.

## 1. Introduction

Recent progress in droplet epitaxy methods has made it possible to grow unique nanostructures, such as quantum rings (QRs) [1,2,3,4]. In particular, QR structures have attracted considerable attention because of the optical Aharonov–Bohm (AB) effect [5,6,7,8], which is analogous to the oscillation of conductance as a function of magnetic flux in an AB interferometer made of a quantum point contact [9]. AB-like oscillations have been observed in charged excitons arising from the motion of a single charge in lithographically defined rings [5,8] and type-II dots [7], but a novel AB effect involving neutral excitons has not been clarified [6,10], wherein stabilized switching between bright and dark exciton states can be realized by the magnetic-field-induced phase.

In general, to observe the AB effect, perfect cylindrical symmetry for rotational motion around an entire ring structure is assumed. The asymmetric shape condition of real QR structures is often ignored. To overcome the anisotropy-induced potential barrier or asymmetrical exchange interaction, strong external magnetic fields are required [11,12,13,14,15,16,17,18,19,20]. In addition, the threshold of the anisotropy-induced potential barrier or asymmetrical exchange interaction is affected by temperature variations. Therefore, the understanding of asymmetry in the ring structure and temperature-dependent dynamics are major factors in investigating the conditions of the optical AB effect. Localized states have been observed in an asymmetric single QR with theoretical calculations of the adiabatic potential [11,12]. In this study, fine structures, strong polarization in the excited state, and the temperature dependence of QRs were investigated with the temperature dependence of quantum dot (QD) structures.

## 2. Materials and Methods

QR structures were grown via the droplet epitaxy method. After deoxidation at 600 °C under As_4_, a 100 nm-thick GaAs buffer layer and a 50-nm thick Al_0_._3_Ga_0_._7_As layer were grown sequentially at substrate temperature (T_s_) of 580 °C. Then, Ga droplets were formed on the Al_0_._3_ Ga_0_._7_As surface at 309 °C, and the droplets were crystallized by As_4_ injection. Diverse parameters for the growth of QRs were varied to observe the change in the surface morphology under each fixed condition. The beam equivalent pressure (BEP) of As_4_ varied in the range of 1.2 × 10^−7^ to 1 × 10^−5^ Torr. The T_s_ varied in the range of 257–321 °C under the given BEP of As_4_. The changes in the QRs were observed with an increase in the interruption time in sequential injection of As_4_ with a BEP of 2.5 × 10^−7^ Torr at T_s_ of 309 °C. The interruption time was in the range of 0–30 s. To form the GaAs QD, the higher As_4_ flux was supplied with BEP of 3.0 × 10^−6^ Torr at the same T_s_ used for a reference QD. The formation mechanisms for the QD and QR are described in our pervious works and in the literature [21,22,23].

The surfaces of the Ga droplets, GaAs QDs, and GaAs QRs were observed by atomic force microscopy (AFM) and scanning electron microscopy (SEM). The optical properties of the grown QRs and QDs were investigated using micro-photoluminescence (PL) systems. The QRs and QDs were capped with 60 nm-thick Al_0_._3_Ga_0_._7_As and 3 nm-thick GaAs for optical measurements. These samples were held in a continuous-flow Janis ST-500 cryostat equipped with a heating resistance so that the range of the accessible temperature was 4–300 K, with a temperature stability of ~50 mK. The cryostat had a support structure that exhibited low-temperature expansion, as well as an internal vibration isolation system to reduce the draft of samples under excitation. A commercial Becker & Hickl SPC-630 single-photon counting card was used, which provided two inputs—one for the excitation laser source and another for the detected PL from the photomultiplier tube (PMT). The synchronization (SYNC) was used to generate electrical pulses from the laser output. A constant fraction discriminator (CFD) provided a logic pulse that was temporally correlated with the detection of a photon at the PMT. This pulse triggered the time-to-amplitude converter (TAC), which generated a linear voltage ramp stopped by a signal from the SYNC photodiode, whose input was the periodic laser signal. An analogue-to-digital-converter (ADC) stored the temporal location of each detected PL photon in the memory. To measure the fast decay dynamics from a single QR/QD structure, a Hamamatsu PMT with 150-ps time resolution and a time-correlated single-photon counting card were used.

## 3. Results & Discussion

The QD and QR structures were examined using atomic force microscopy (AFM) and scanning electron microscopy (SEM) images of the uncapped sample, respectively, as shown in Figure 1a. The cross-sectional profiles of the QRs were measured by AFM. We investigated over 40 QDs and QRs to estimate their average sizes. The average lateral size and height of the QDs were 14 ± 1 nm and 10 ± 1 nm, respectively. The actual QD density was 3 × 10^9^/cm^2^. The actual QR density was 2 × 10^9^/cm^2^. The average lateral sizes of the QR along the [110] and the [1$\bar{1}$0], direction were 17 ± 1 nm and 25 ± 1 nm, respectively. The size of the QR along the [110] direction was relatively small compared to that along the [1$\bar{1}$0] direction. This indicates that the Ga migration length along the [1$\bar{1}$0] direction was much longer than that along the [110] direction at the given growth conditions. The average height of QR was approximately 10 ± 1 nm. 

The PL of a single QR/QD was collected at 4 K using a confocal arrangement, where frequency-doubled (400 nm) Ti/sapphire laser pulses (120 fs pulse duration at an 80 MHz repetition rate) were focused on the QR/QD sample (~7 QRs/µm^2^) with a spot size of 0.8 µm^2^. Approximately five single structures with a spot size of 0.8 µm^2^ were excited, and the main issue of the micro-PL measurements was the regularity of the samples, e.g., their shapes and sizes. Through careful investigation of the nanostructures, we confirmed their regularity, including distribution issues for micro-PL measurements. A time-correlated single-photon counting system was used to obtain the time-resolved PL.

As shown in Figure 1b, two additional PL peaks emerged at lower (1.721 eV) and higher (1.738 eV) energy levels relative to the ground-state exciton (1.727 eV) as the excitation power was increased at 4 K under spectral resolution ~20 μeV from a single QR sample (the excitation power increased from 1 to 4 kW/cm^2^). The excess energy (11 meV) of the higher-energy peak is comparable to the theoretical value (13 meV) of the excited exciton state (*N* = 2) in a single GaAs QR, denoted by the radial principal quantum number (*N*) [1,2]. The lower-energy peak was likely due to biexciton (XX) emission, characterized by a rapid rise relative to the exciton peak. 

The high biexciton binding energy (~6 meV) in the QR (relative to a few millielectronvolts in a GaAs QD [18]) is attributed to the strong confinement. Even though the QR has an asymmetric structure, as long as it maintains approximately cylindrical symmetry, the exciton fine-structure states (N, L) can be specified in terms of the additional azimuthal quantum number (L), where the angular momentum of the rotational motion around a ring is quantized. Supposing that the ring width is negligible, the level spacing of the fine-structure states can be estimated as $E_{Le,h}={\left(\hbar L_{e,h}\right)}^{2}/\left(2\times m_{e,h}^{*}\right)$.Given the average ring radius (R ~ 20 nm) obtained from the SEM image and the effective masses of the electron (e) and heavy hole (h) in GaAs [2] ($m_{e,h}^{*}$ = 0.067 m_0_ and $m_{h}^{*}$ = 0.51 m_0_, where m_0_ represents the electron rest mass), as many as seven degenerate fine-structure levels (*L*_e,h_ = 0, ±1, ±2, ±3) are possible within the PL linewidth (~1.5 meV) of the X (*N* = 1) and X (*N* = 2) states. 

As the excitation power was increased, these fine structure states from X (*N* = 1) and X (*N* = 2) were filled sequentially, giving rise to a blue shift, as shown in Figure 1b. Meanwhile, the exciton PL peak from a single QD showed no shift with increasing excitation power, as seen in Figure 1b. In Figure 1c, the PL spectrum can be seen to exhibit a strong polarization dependence for the X (*N* = 2) state measured at 2 kWcm^−2^ at 4 K. In general, the spectrometers showed a dependence on the polarization of the signal from the samples. However, this polarization effect was controlled with setup alignments, and no polarization effects were observed when symmetric test samples were used. According to these results, we performed polarization-dependent measurements of the QD and QR structures. The PL spectrum exhibited a redshift as the analyzer angle increased from 0° to 90°, but the PL intensity decreased significantly. Moreover, the analyzer angle dependence of the PL intensity was mapped at different energies in polar coordinates, where the maximum intensity of each PL spectrum was normalized for comparison. 

In the case of an elliptical QD, an asymmetric electron-hole exchange interaction leads to a splitting of the double exciton states (|±1>) into two singlet states (|X,Y> = (|+1> ± |−1)/√2), where two linearly and orthogonally polarized dipoles (|X> and |Y>) are defined along the principal axes of an elliptical QD [13,14]. This gives rise to the energy splittings, which involves the transition from biexciton to two singlet exciton states (|X,Y>), respectively. A *k*=3 localized state of the exciton under an adiabatic potential $\varepsilon_{e,h}^{k}$ (r, ϕ), with strong polarization dependence at the X (*N* = 1) and XX states to support the *k* = 3 localized state, was reported in an asymmetric single QR [11,12] The relative energy difference (∆) of 1.14 meV in the X (*N* = 2) state was also substantial. This result is attributed to the breaking of the significant selection rule in the *k* = 3 localized state of the lower-symmetry QR structure.

As the temperature increased, the PL intensities in the *X* (*N* = 1) and *X* (*N* = 2) states decreased with linewidth broadening. Strong exciton–phonon interaction was observed in the *X* (*N* = 2) state. Because of the strong exciton–phonon interaction, the *X* (*N* = 2) state exhibited a larger redshift and stronger linewidth broadening than the *X* (*N* = 1) state. To confirm this strong exciton–phonon interaction in the *X* (*N* = 2) state, the linewidth broadening and bandgap were measured as the temperature increased, as shown in Figure 2a. Here, Equations (1) and (2) were used to fit the linewidth and bandgap change [24,25].
(1)$$A\left(T\right)-A\left(0\right)=S_{ac}T+S_{op}\left(1/\exp\left(\hbar \omega_{LO}/k_{B}T\right)-1\right)$$

Here, *S*_ac_ is the exciton–acoustic phonon interaction coefficient, *S*_op_ is the exciton–optical phonon interaction coefficient, *ħω_LO_* is LO-phonon energy and *k_B_* is the Boltzmann constant. The first term shows the acoustic scattering, while the second term gives the scattering with optical phonons. At low temperature, *k_B_T* became much smaller than ħω_LO_ (*k_B_T* << *ħω_LO_*). The second scattering channel can be ignored due to the negligible LO-phonon population. On the other hand, at high temperature, the first channel can be neglected due to *S_ac_* << *S_op_*. The linewidth broadening can be well fitted by Equation (1) with the acoustic phonon interaction coefficient (14 µeV/K) and optical phonon interaction coefficient (120 meV) for the ground exciton (*N* = 1) state, and the acoustic phonon interaction coefficient (14 µeV/K) and optical phonon interaction coefficient (245 meV) for the excited exciton (*N* = 2) state.
(2)$$E_{g}\left(T\right)=E_{0}-2a_{b}/\left(\exp\left(\theta_{b}/T\right)-1\right)$$

Under the Bose–Einstein model given by Equation (2), *E*_0_ represents the bandgap energy at 0 K, *a_B_* represents the strength of the exciton average phonon (optical and acoustic) interaction [25], and the fitting parameter $\theta_{B}$ represents the average temperature of the acoustic and optical phonons. In Figure 2a, the obtained fitting parameters are the bandgap energy (*E*_0_) values at 0 K, i.e., 1.727 eV for the X (*N* = 1) state and 1.738 eV for the X (*N* = 2) state, and *a_B_* which is connected by the strength of the exciton–average phonon: 27 meV for the X (*N* = 1) state and 48 meV for the X (*N* = 2) state. A stronger exciton–phonon interaction was observed in the X (*N* = 2) state than in the X (*N* = 1) state, as shown in Figure 2a. In Figure 2b, the blue line indicates division of the integrated area of the X (*N* = 2) energy peak by the integrated area of the X (*N* = 1) energy peak. Because of the strong exciton–phonon interaction in the X (*N* = 2) state, the relative integrated area ratio (X (*N* = 2)/X (*N* = 1)) increased continuously depending on the temperature increase. This ratio increase indicates that the PL intensity and linewidth of the X (*N* = 2) state were affected by the stronger exciton–phonon interaction.

To compare the confined electron activation energies of the GaAs QD and GaAs QR structures, PL measurements were performed as a function of the temperature. As the temperature increased, the intensities of the PL spectra decreased gradually, and a nonradiative decay process became dominant; therefore, PL quenching at high temperatures can be attributed to carrier dynamics in the context of a nonradiative recombination process. Because of the decrement in the energy bandgap between the electronic ground state and the heavy-hole ground state, the PL peaks shifted to lower energies. As the temperature increases, the integrated PL intensities can be explained by
(3)$$I=I_{0}/\left[1+C_{\exp}\left(-\Delta E_{A}/k_{b}T\right)\right]$$
where *I*_0_ represents the integrated PL intensity at 0 K, C represents the ratio of the thermal escape rate to the radiation recombination rate, and Δ*E_A_* represents the active tion energy for the barrier height of the nonradiative recombination channel [26].

To determine and compare the activation energies of confined electrons, we performed temperature-dependent PL measurements, as shown in Figure 3a,b. The observed activation energies of confined electrons in GaAs QRs and GaAs QDs from the integrated PL intensities as a function of the temperature were 86 and 66 meV with error bars, respectively. The activation energy of the GaAs QRs exceeded that of the GaAs QDs. This is attributed to the enhanced confinement resulting from the unique QR structures [27], which induced stronger confinement states as the quasi-one-dimensional density of states, as long as the confined exciton was permitted to rotate around the entire ring structure with a finite width [14,28]. A QR is composed of two inner and outer ellipses, and the ring width is determined by the relative length between the inner and outer ellipses. Therefore, the confinement effect is more dominant in a QR than in a QD, because of the smaller effective area, although the overall sizes of a single QD and a single QR are comparable.

We performed time-resolved PL measurements to investigate the carrier dynamics of a single QD and a single QR. As shown in Figure 4, the radiative decay time of a single QR structure was shorter than that of a single QD structure. In general, the radiative decay time decreased with a reduction in the confinement size, due to the increased electron–hole wavefunction overlap with the increasing exciton oscillator strength in small confinement areas [29,30,31]. The obtained time-resolved PL spectra were characterized by single exponential decay, which was used to extract the radiative recombination time [32] for a single QD (~750 ps) and a single QR (~580 ps). Compared with the case of a single QD, the confinement effect of a single QR was stronger owing to the asymmetric ring structure, resulting in more confined motion of the exciton states. The shorter exciton decay time of a single QR is attributed to the increased trap barrier energy, resulting in strongly confined electrons in the QR.

## 4. Conclusions

Fine structures were observed from ground (*N* = 1) and excited (*N* = 2) state excitons in power-dependent PL measurements at 4 K. The strong polarization dependence of the excited exciton state was confirmed with a large energy separation between analyzer angle rotations. The activation energy of the confined electrons in a single QR (86 meV) was approximately 20 meV higher than that in a single QD (66 meV), due to a strong confinement effect. The effective confinement area was determined by the relative distance between the inner- and outer-ring ellipse structures.

## Figures and Tables

**Figure 1 nanomaterials-12-02331-f001:**
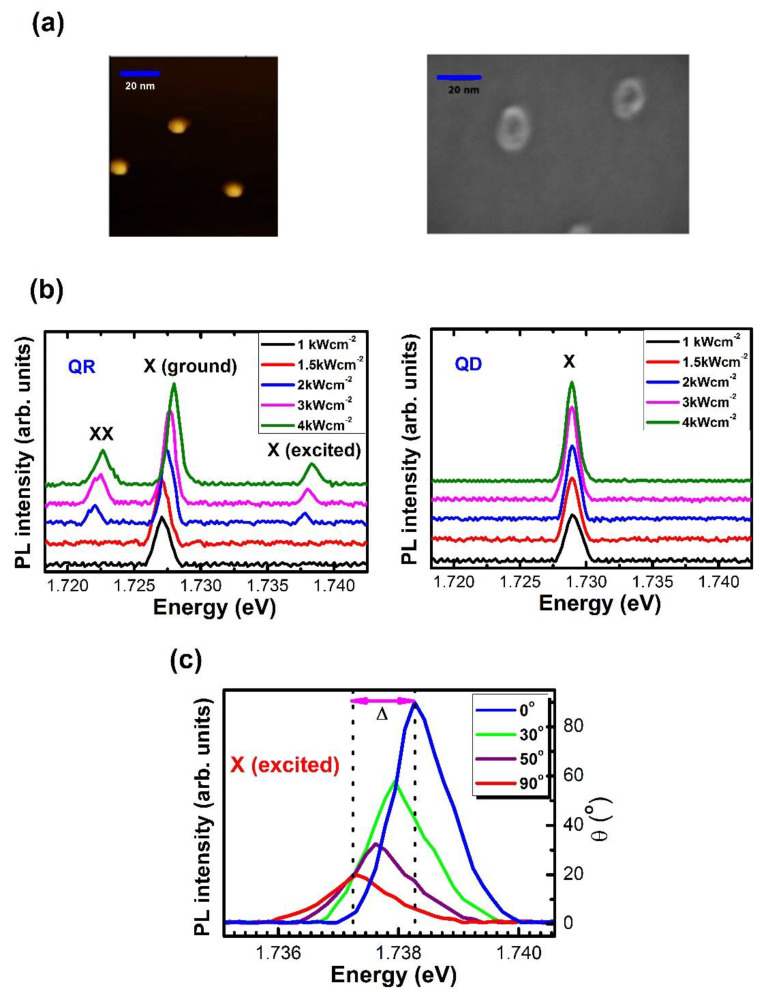
(**a**) Single QD/QR structure observed via SEM. (**b**) Excited X (*N* = 2) and XX (biexciton) states appear from a single GaAs QR with increasing excitation power. The ground- and excited-state exciton (X) is blue-shifted because of the sequential filling of fine-structure states. Only ground exciton states are observed from a single QD with no peak shifts. (**c**) PL spectra of X (*N* = 2) at different analyzer angles, exhibiting strong polarization dependence between the analyzer angles of 0° and 90° with energy splittings (Δ).

**Figure 2 nanomaterials-12-02331-f002:**
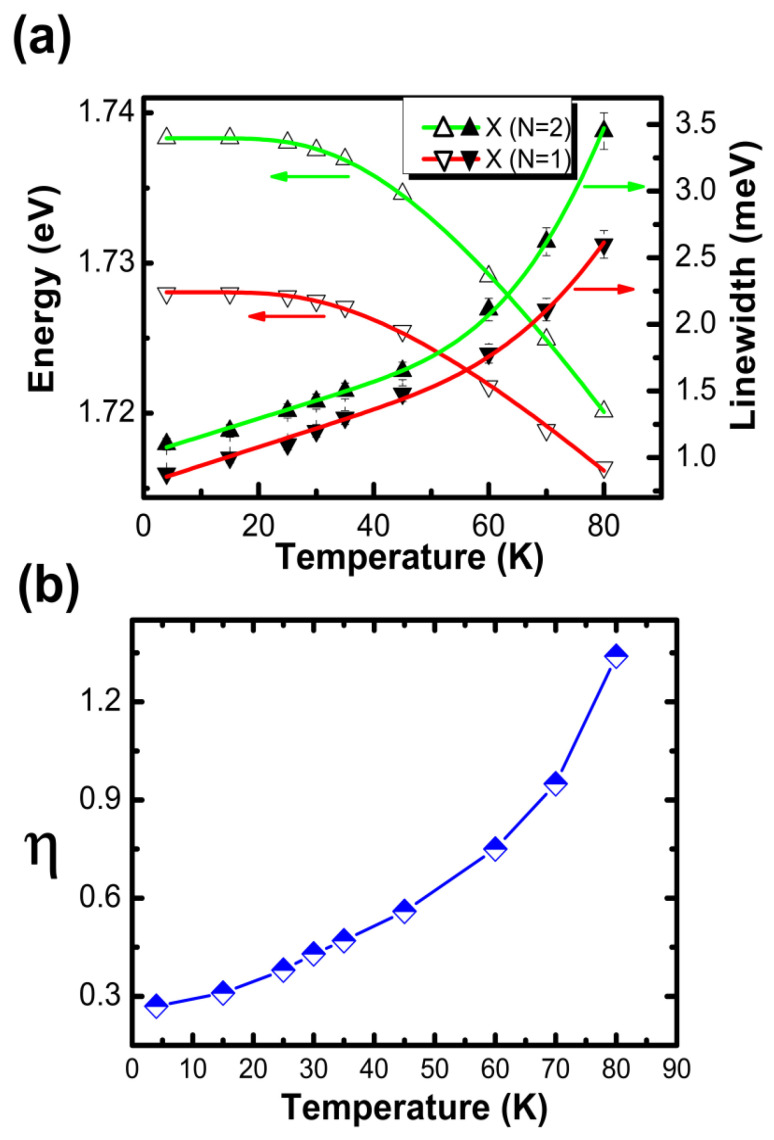
(**a**) Bandgap shift and linewidth broadening for the temperature change with fitting via the Bose–Einstein model and the exciton–phonon interaction formula at X (*N* = 1) and X (*N* = 2). (**b**) Integrated area ratio of the X (*N* = 1) PL peak to the X (*N* = 2) PL peak as a function of the temperature.

**Figure 3 nanomaterials-12-02331-f003:**
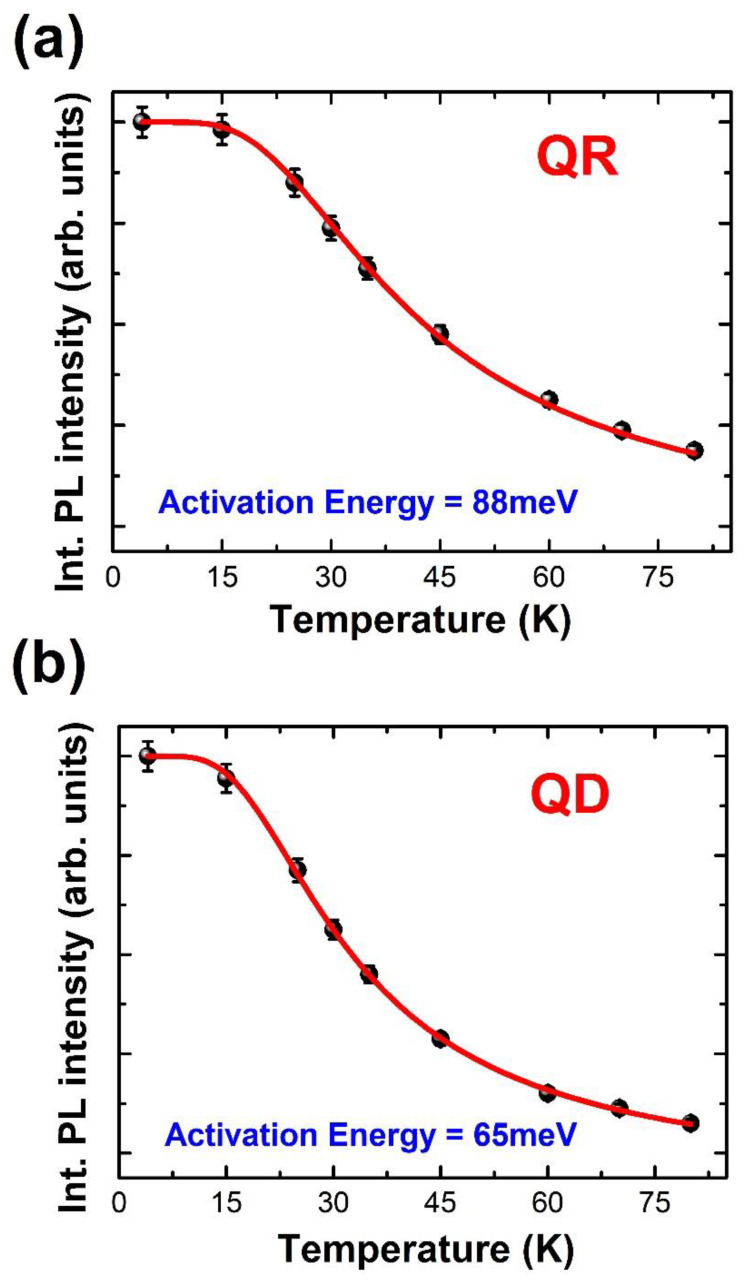
Integrated PL intensities with respect to the temperature for the (**a**) QR and (**b**) QD.

**Figure 4 nanomaterials-12-02331-f004:**
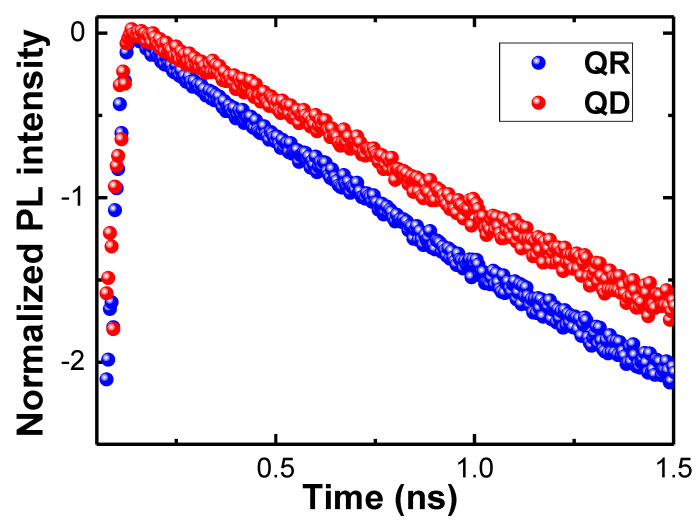
Time-resolved PL spectra at 4 K for comparing the carrier dynamics of the QR and QD. The PL decay curves for the two structures exhibit different decay behaviors.

## Data Availability

Not applicable.

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
