# Peer review of "Temperature-Dependent Exciton Dynamics in a Single GaAs Quantum Ring and a Quantum Dot"

_nanomaterials, 2022, doi:10.3390/nano12142331_

Round 1

Reviewer 1 Report

  1. The article title weakly correlates with the content. The article does not disclose the description of GaAs quantum dots used for comparison, their structural (morphological) and optical characteristics. In fact, the authors conduct comparative studies without providing data on the reference system.

  1. In section 2 "Materials and Methods", the authors present the ranges of modes for manufacturing nanostructures (droplets, quantum rings and quantum dots). Why is this information given if the authors do not provide further data on the obtained structures? In fact, the entire article is based on the study of only two samples, and structural information about the second sample (with quantum dots) is completely absent.

  1. What are the two types of Ga droplets in question? And why this information if it is not used in the future? And why are we talking about droplets if the article title contains quantum dots.

  1. It is not entirely clear how the authors measured the PL from a single nanostructure with a specified laser spot area of 0.8 µm2 and surface density of the structures of 7 µm-2. In this case, at least 5 structures were excited at once. In this regard, questions arise regarding the results of studies by the photoluminescence method.

The article does not correspond to the level of this journal. In fact, standard studies of the optical properties of a single standard sample obtained by a standard method for droplet epitaxy are carried out. Moreover, the results of these studies also have several questions, as shown above. In fact, there is no coherent and structured design of study. In this regard, I am forced to recommend that the article be rejected.

Reviewer 2 Report

Well done work overall. I have a few minor comments and one relatively major concern (reason for which I did not answer on the scientific soundness). My major concern is related to the dependence of the signal with the analyzer angle. Spectrometers have a relatively strong dependence on the polarization of the light analyzed, what were the corrections or the compensation methods utilized to make sure that did not affect the results obtained? This should be described in the article.

As for the minor comments, the part describing TC-SPC detection (lines 63 to 69) is somewhat confusing and could be significantly improved. Just by reading that paragraph, if I didn't know how the technique works, I wouldn't be able to understand it. Please rewrite it.

Line 132 has an extra "is", delete it.

Finally, the time scale in figure 4 is wrong, please correct it.

Reviewer 3 Report

Kim et al. present intensity varied photoluminescence data and temperature-dependent measurements of the exciton dynamics of GaAs rings compared to dots. The overall results look promising, but the manuscript is in poor condition, as this brief review will exemplarily show. As a result, my recommendation is major revision or rejection due to ill-documented presentation of scientific results. In particular, important detailed information are not given, which makes it difficult for the reader to follow the main rationale of this contribution. The following list is not complete, but the manuscript as a whole needs careful revision.

Chapter 2, Materials and Methods: Temperature dependence is an important part of the manuscript. However, this subchapter lacks a proper description of temperature control and accuracy of the measurements. Please add. In addition, I could not find any information on the shape, size distribution and quality of the quantum dots used in these studies. How many quantum dots were simultaneously illuminated with the femtosecond laser (the corresponding number is 7 for the quantum rings as highlighted in line 77)?

Chapter 2/3, question for understanding: I did not understand where the spectral resolution in the photoluminescence measurements come from if only a photomultiplier and a single-photon counter (lines 70 and 71) were used. These devices including the Becker & Hickl SPC-630 do not give spectral resolution as implied in lines 79 and 80. What was the spectral resolution of the measurements? Please explain in the main text because this is important for the basic understanding of the results.

Line 155: The definition of the Boltzmann constant needs to be shifted to the description of equation (1). In line 155, it is too late.

The missing description of the quantum dots brings me to another major concern. Maybe the differences between ring and dot structures are only due to structural differences and could also occur within quantum dots of different average sizes. Again, the question arises what is the accuracy of the measurements. What are typical variations in size and shape of the quantum rings and dots? Did the authors measure at several different spots or only at one (question of reproducibility)? How does shape and size influence optical properties and temperature dependence? Do all 7 quantum rings illuminated by the femtosecond laser (line 77) have the same diameter or is there a distribution as well?

Figure 1b and c: A color code is missing in the figure caption. There is one in Figure 1b, but not in the underlying text.

Background information (referencing) of equations 1 to 3 is needed. What does the subscript “LO” mean? In its present form, equation (2) appears to be unphysical because of different units of 1 (dimensional) and E_0 (units of energy). Is there a typo (missing bracket)?

Figure 3: Please add error bars for the determination of the activation barriers. Is a difference of 17 meV significant? Please comment.

Figure 4 and the corresponding text starting at line 169: In the text, decay constants in the range of hundreds of picoseconds are mentioned, but the abscissa axis in Figure 4 reads seconds only.

The discussion section for figures 3 and 4 (lines 156 to 178) is of poor quality. Almost no references are given, only claims/speculations without the possibility to verify them.

References: There are only 20 references listed. With regard to the topic, this is a very low number. Please improve. Ideas how to do that are given above in this review.

Round 2

Reviewer 1 Report

The authors made changes to the article, but in general its format has not changed.

From the revised description of the Materials and Methods section, it is still not clear what the structure of the samples for PL was, or whether the PL studies were carried out directly on the surface structures.

In addition, it is not clear how SEM differs from FESEM as far as I understand, they are one and the same.

In Figure 1a (left), the authors show the QD image and indicate that this is a SEM image. However, it is obvious that we have an AFM scan. In my opinion, this is completely unacceptable. It is necessary to provide either SEM images or AFM images for both types of structures.

There are no data on the spectra of quantum dots. And this is strange, given the title of the article. Quantum dots appear only at the end.

There is no deep analysis of the causes of polarization effects in ring structures. Otherwise, the article reproduces well known and obvious facts.

My opinion remains unchanged this material does not correspond to the level of the journal.

Reviewer 3 Report

see attachment
